# COVID-19 Peritraumatic Distress and Loneliness in Chinese Residents in North America: The Role of Contraction Worry

**DOI:** 10.3390/ijerph19137639

**Published:** 2022-06-22

**Authors:** Andrea D. Y. Lee, Peizhong Peter Wang, Weiguo Zhang, Lixia Yang

**Affiliations:** 1Department of Psychology, Toronto Metropolitan University, Toronto, ON M5B 2K3, Canada; dya.lee@ryerson.ca; 2Division of Community Health and Humanities, Faculty of Medicine, Memorial University of Newfoundland, St. John’s, NL A1B 3V6, Canada; pwang@mun.ca; 3Dalla Lana School of Public Health, University of Toronto, Toronto, ON M5T 3M7, Canada; 4Centre of New Immigrant Well-Being (CNIW), Toronto, ON L3R 3M8, Canada; 5Department of Sociology, University of Toronto Mississauga, Mississauga, ON L5L 1C6, Canada; weiguo.zhang@utoronto.ca

**Keywords:** COVID-19 peritraumatic distress, loneliness, contraction worry, Chinese residents, COVID-19 pandemic

## Abstract

The current study examined the association of COVID-19 contraction worry for self and for family members with COVID-19 peritraumatic distress and loneliness in Chinese residents in North America. A sample of 943 Chinese residents (immigrants, citizens, visitors, and international students) in North America completed a cross-sectional online survey during the second wave of the COVID-19 pandemic (between January and February 2021). Univariate analysis of variance (ANOVA) models identified possible sociodemographic variables that were included in the subsequent hierarchical regression models. According to the hierarchical regression models, self-contraction worry was significantly associated with both COVID-19 peritraumatic distress (B = −4.340, *p* < 0.001) and loneliness (B = −0.771, *p* = 0.006) after controlling for related sociodemographic covariates; however, family-contraction worry was not significantly associated with the outcome variables. Additionally, poorer health status and experienced discrimination significantly predicted higher COVID-19 peritraumatic distress, whereas poorer health status and perceived discrimination significantly predicted increased loneliness. The results highlighted the detrimental impacts of self-contraction worry on peritraumatic distress and loneliness during the COVID-19 pandemic in Chinese residents in North America.

## 1. Introduction

Since the World Health Organization declared COVID-19 as a pandemic [1], the virus and the related restriction measures such as lockdown and social distancing have resulted in adverse psychological impacts [2,3,4,5]. Previous research has reported an increase in depression and anxiety symptoms, as well as worry and stress during the pandemic [6,7,8]. Past research found that worry was positively associated with peritraumatic distress in populations in Israel [9], France [10], and South Korea [11], and worry was positively associated with loneliness in Norway [12] and the United States [13]. However, to our knowledge, little is known about whether and how worry is related to COVID-19 peritraumatic distress and loneliness among Chinese residents in North America, an especially vulnerable population considering the rising anti-Asian racism and discrimination during the pandemic [14,15]. To fill this gap, the current study aimed to specifically examine contraction worry (i.e., worry about self or family members’ contraction of the COVID-19 virus) as a risk factor for COVID-19 peritraumatic distress and loneliness in Chinese residents (i.e., immigrants, citizens, visitors, students, etc.) in North America.

### 1.1. Worry

Worry refers to the relatively uncontrollable occurrence of repeated ideas, thoughts, and images about uncertain and possibly adverse events [16,17]. As a fundamental cognitive feature of anxiety, worry negatively impacts well-being and mental health [18]. During the COVID-19 pandemic, it has been found that worry about health, financial consequences, and economic and political impacts of the pandemic was associated with lower life satisfaction, decreased self-rated health, increased loneliness, and higher psychological distress [19,20]. In addition, a higher level of reported worry about COVID-19 was significantly associated with depression, anxiety, and stress [13]. Additionally, worry about the dangerousness of COVID-19 was central to the COVID Stress Syndrome and related avoidance and self-protective behaviors [21].

The COVID-19 pandemic has elicited higher levels of fear and worry, coined “Coronaphobia” (i.e., anxiety felt toward contracting the virus) [22]. In a qualitative study, researchers identified fear of virus contraction or transmission as one of the major themes in COVID-19 distress among university students [23]. This contraction anxiety was especially directed toward family members who were at a higher risk of contraction. A heightened COVID-19 contraction risk perception was positively related with greater worry about COVID-19 and thus increased anxiety, depression, and negative mood [24]. Additionally, anxiety regarding contracting COVID-19 was positively associated with anxiety sensitivity [25], and concerns about contagion of COVID-19 for self and family were associated with an increased negative affect [26]. In light of these findings, the present study specifically examined the impact of viral contraction worry as a risk predictor for peritraumatic distress and loneliness in Chinese residents in North America.

### 1.2. Peritraumatic Distress

Peritraumatic distress refers to cognitive and emotional distress throughout and immediately following a traumatic event [27]. It is positively associated with post-traumatic stress disorder (PTSD) and other psychiatric outcomes such as anxiety, depression, sleep disturbances, and traumatic grief [28]. The COVID-19 pandemic can be considered a traumatic stressor, because although the pandemic is not listed as a trauma in the DSM-5, the COVID-19 pandemic has been shown to elicit PTSD-like symptoms and high levels of peritraumatic distress [11,29]. The COVID-19 pandemic has many of the characteristics of a mass traumatic event [30], and it has been shown to exacerbate other related mental health problems such as depression, anxiety, and psychosocial functioning [29]. Specifically, peritraumatic distress during COVID-19 also significantly predicted post-traumatic stress symptoms, generalized anxiety, and depression symptoms in all age groups [10]. Similarly, peritraumatic distress was associated with higher anxiety symptoms and lower resilience [31]. Therefore, there is a need for more research into the traumatic effect of the COVID-19 pandemic [30].

Peritraumatic distress has been associated with different risk factors during the COVID-19 pandemic, such as poor health, increased media use [11,31], increased levels of health worry [9], and increased levels of worry about the COVID-19 pandemic [10]. In a study conducted in South Korea, the perception of risk and fear of COVID-19 was associated with elevated levels of peritraumatic distress, negative emotions, and life threat [11].

### 1.3. Loneliness

Loneliness is the discrepancy between desired and actual levels of social relationships [32,33]. It is the unpleasant state when an individual senses that their social needs are not met and notices a difference between the desired and actual amount and quality of social interaction available in the individual’s environment [34]. Loneliness is associated with physical, psychiatric, and psychosocial risk factors such as depression, alcoholism, suicidal thoughts, social anxiety, obesity, and decreased immunity [12,34]. It is more likely to occur in populations who are at risk for social alienation, isolation, and separation [34].

During the COVID-19 pandemic, the rise of loneliness has been of specific concern, as it is one of the negative consequences of social distancing and quarantine measures, such as physical distancing from close individuals, feeling alienated, and hindered access to coping mechanisms for loneliness, such as social activities [35]. In fact, studies conducted during the pandemic have suggested that there has been an increase in loneliness [36], and higher reported levels of loneliness were associated with extended exposure to the pandemic [37]. In studies examining loneliness during COVID-19, people living alone, younger adults, people with chronic illness, and economically inactive individuals reported a higher level of loneliness compared to their counterpart groups [38,39]. Older Chinese adults in Canada may experience loneliness and social isolation due to risk factors such as intergenerational tension, living alone, language barriers, and limited social networks beyond family [40]. This is alarming because loneliness in family relationships is associated with worry, and loneliness in social relationships is associated with anxiety and depressive symptoms [41].

When Canada first initiated the stay-at-home order in March 2020, strict lockdown measures were implemented in response to the rapid spread of the virus, and these were found to be associated with greater health anxiety, financial worry, and loneliness [5]. Older adults identified confinement, restriction, concern for others, isolation, and loneliness as the most stressful during the pandemic [42]. In Norway, it was found that increased rumination and worry were associated with stronger loneliness [12]. Moreover, worry regarding COVID-19 was positively correlated with loneliness in college students, and the interaction between worry and loneliness was significantly associated with depression, anxiety, and stress [13].

### 1.4. Chinese Residents and COVID-19

Chinese residents have experienced unique mental health challenges during the COVID-19 pandemic [43,44]. Increased anti-Asian racism, feelings of double-unbelonging with regard to both host country and China, and disapproval of political criticism targeting China may pose challenges for Chinese oversea residents due to their cultural, social, and political ties to China [43]. Asian individuals in the United States and France have reported increased racially motivated hate crimes involving physical violence and harassment since the COVID-19 pandemic began [45,46]. In France, Chinese residents with French citizenship reported greater discrimination compared to Chinese residents without citizenship, and the experience of discrimination was exacerbated by media sources using terms such as “Chinese virus” or “Yellow Alert” [46]. It has brought to light that Asian immigrants have been treated as “forever foreigners” [47]. In the Netherlands, Chinese immigrants reported decreased mental health conditions associated with the rise of racism, social isolation, and distrust in Dutch COVID-19 information [48]. Furthermore, fear of the virus was significantly correlated with social isolation and racism in Chinese immigrants [48]. In this context, the current study focused on Chinese adults living in North America while considering their well-reported increased risk for negative psychosocial outcomes from COVID-19 [44].

### 1.5. Present Study

The present study aimed to examine the predictions of self- and family-contraction worry for peritraumatic distress and loneliness in Chinese residents in North America. Previous research suggested that concern about the contagion of COVID-19 was associated with greater anxiety and negative affect [24,25,26]. To our knowledge, there has been a lack of studies examining COVID-19 contraction worry in Chinese residents in North America. Additionally, no studies have considered contraction worry for self as well as for immediate family members. Given the collectivistic cultural orientation in China [49], we hypothesized that both self-contraction and family-contraction worry in this sample would predict COVID-19 peritraumatic distress and loneliness. Thus, we assessed both types of worry in the current study.

The main purpose of the current study was to examine whether the worry about contracting COVID-19 for self or family would predict peritraumatic distress and loneliness in Chinese residents in North America. In light of the literature, we hypothesized that: (1) higher self-reported self- and family-contraction worry would predict higher COVID-19 peritraumatic distress; and (2) higher self-reported self- and family-contraction worry would predict higher levels of loneliness, even when controlling for related sociodemographic covariates.

## 2. Materials and Methods

The study received ethics approval from Ryerson University (now Toronto Metropolitan University) (REB 2020-132) and Memorial University of Newfoundland (20201772-ME).

### 2.1. Sample

This study recruited Chinese residents living in North America, including Canadian and American citizens and permanent residents of Chinese ancestry, Chinese international students, and people who held valid work permits or visitors. The participants were recruited by distributing the survey link and QR code in a flyer or news report on media platforms (e.g., WeChat groups, WeChat official forums, emails, websites, TV or radio reports, and the Google search engine). Participation was voluntary, and informed consent was obtained before the start of participation. All ineligible participants who did not meet the following inclusion criteria were excluded from analysis: (1) Chinese residents who had lived in North America (i.e., Canada and the United States) for at least 6 months; and (2) aged 16 years or older. A total of 1214 participants attempted to complete the survey. After removing incomplete and ineligible cases (*n* = 271), the final sample consisted of 943 participants. A majority of the participants were female (64.5%), ages 35–64 (69.2%), married or partnered (82.2%), and living with family members (86.1%). Sample characteristics can be found in Table 1.

### 2.2. Measures

The survey was built in Qualtrics^TM^ and delivered in simplified or traditional Chinese. The cross-sectional survey was distributed in the second year of the COVID-19 pandemic in North America (i.e., from 11 January to 27 February 2021). All identifying information, such as IP and email address, were removed before analysis. The survey included measures of COVID-19 peritraumatic distress, loneliness, worry, and demographic variables. Peritraumatic distress and loneliness served as the main outcome measures, whereas self- and family-contraction worry were the primary predictors. Related demographic variables were included as covariates.

#### 2.2.1. The COVID-19 Peritraumatic Distress Index (CPDI) Scale

Peritraumatic distress was measured using the CPDI scale, a well-reported and validated measure widely used during the COVID-19 pandemic [50,51,52,53]. It was originally developed to evaluate COVID-19 peritraumatic distress in China, including 24 items assessing the frequency of anxiety, depression, phobia, cognitive change, avoidance and compulsive behavior, physical symptoms, and loss of social functioning in the past week [53]. An example item is: “I feel helpless and angry about people around me, government, and media” [52]. Participants were asked to rate each item using a 5-point Likert scale ranging from 0 (never) to 4 (very often), resulting in a sum score that ranged from 0 to 96. A sum score of 28–51 indicated mild to moderate distress, and a score of 52 and greater indexed severe distress [53]. The CPDI scale was used among Chinese, Spanish, and Italian samples, with a high reliability (Cronbach’s a = 0.92–0.95) [51,52,53]. The current study also confirmed a high reliability (Cronbach’s a = 0.94). For the purpose of this study, the CPDI scale was used to measure the peritraumatic distress during the COVID-19 outbreak, with a higher score indicating a higher level of distress.

#### 2.2.2. The 6-Item De-Jong Gierveld Loneliness Scale

Loneliness was measured with a modified version of the 6-item De-Jong Gierveld Loneliness Scale [54], a valid and reliable measure of overall, emotional, and social loneliness. In the current study, the items were modified to add the context of the pandemic by including the phrase “during the pandemic” in each item. Participants were asked to rate each item (e.g., “I experience a general sense of emptiness during the pandemic”) based on a 5-point Likert scale ranging from 1 (strongly agree) to 5 (strongly disagree) or vice versa (for negatively phrased items). In this study, the total score was used as a measure of loneliness, with higher scores indicating elevated levels of loneliness (with negatively phrased items reversely coded first). The internal consistency of this modified COVID-specific loneliness scale was satisfactory (Cronbach’s a = 0.72).

#### 2.2.3. Contraction Worry

Contraction worry was measured with two questions: one assessing self-contraction worry (i.e., “Do you worry about yourself contracting the COVID-19?”) and the other assessing family-contraction worry (i.e., “Do you worry about someone in your immediate family contracting the COVID-19?”). For both items, participants were asked to rate based on a scale ranging from 1 (very much) to 5 (not at all). In this study, the two items were significantly correlated (*r* = 0.751, *p* < 0.01) and highly consistent (Cronbach’s a = 0.86), with lower scores reported on these items indicating higher levels of worry.

#### 2.2.4. Sociodemographic Variables

Sociodemographic variables included such variables as age, gender, education level, marital status, employment status, length living in Canada, living arrangements, whether they lived with children or pets, religion, health status, and current financial satisfaction. Additionally, we also assessed experienced and perceived anti-Chinese discrimination related to COVID-19 pandemic.

### 2.3. Statistical Analyses

The data analysis was conducted in IBM SPSS 24.0. For clarity purposes, the sociodemographic variables were recoded into binary or 3-level categorical variables based on the outcome variable distributions across each of these sociodemographic variables. The univariate ANOVA models were conducted using the outcome variables (i.e., COVID-19 peritraumatic distress and loneliness) to identify possibly significant sociodemographic covariates, using a criterion of *p* ≤ 0.10. This was stricter than the cut-off of 0.20 found in the literature [55] to keep our focus specifically on the main predictive variable, contraction worry, by limiting the covariates to the most likely sociodemographic predictors. Subsequently, two 2-step hierarchical regression models were conducted, one for COVID-19 peritraumatic distress and the other for loneliness. In Step 1, we entered all the sociodemographic covariates identified in the univariate ANOVAs were entered. In Step 2, we added self- and family-contraction worry as primary predictors. Missing data points were removed with a pairwise deletion approach.

## 3. Results

### 3.1. The Effects of Sociodemographic Variables on the Outcome Variables

The effects of the sociodemographic variables on the outcome variables (i.e., COVID-19 peritraumatic distress and loneliness) were examined using the univariate ANOVAs (see Table 1). The results identified the following sociodemographic variables as potential risk factors (i.e., covariates) for COVID-19 peritraumatic distress: gender, age, marital status, highest education, living arrangement, housing, years in North America, financial satisfaction, health status, weight change, experienced discrimination, and perceived discrimination (*p*s = 0.000–0.083). The following potential sociodemographic covariates for loneliness: gender, marriage status, highest education, living arrangement, housing, financial satisfaction, health status, weight change, experienced discrimination, and perceived discrimination (*p*s = 0.000–0.061). Table 1 presents the *F* and *p*-values for these analyses.

### 3.2. Contraction Worry and COVID-19 Peritraumatic Distress

Self-contraction and family-contraction worry were significantly correlated with COVID-19 peritraumatic distress (*r* = −0.479 and *r* = −0.421, respectively (*p*s < 0.001)). Table 2 presents the results of the hierarchical regression model on COVID-19 peritraumatic distress, with self- and family-contraction worry as the primary predictors controlling for potential sociodemographic covariates. Step 1, with all identified sociodemographic covariates, resulted in an *R*^2^ = 0.244, *F* (18, 309) = 5.526, and *p* < 0.001. Step 2, with self-contraction worry and family-contraction worry added, resulted in *R*^2^ = 0.372, *F* (2, 307) = 9.110, and *p* < 0.001. The difference between Step 1 and Step 2 was Δ*R*^2^ = 0.129, *p* < 0.001, suggesting that contraction worry explained a significant proportion of variance in COVID-19 peritraumatic distress when controlling for all the potential sociodemographic predictors. Specifically, self-contraction worry was identified as a significant predictor [B = −4.340, *p* < 0.001, 95% CI (−6.480, −2.200)], but family-contraction worry was not [B = −1.973, *p* = 0.067, 95% CI (−4.082, 0.137)]. A 1-point increase in self-contraction worry predicted an increase of 4.340 points in COVID-19 peritraumatic distress.

Among the sociodemographic covariates, poorer health status [B = −5.329, *p* < 0.001, 95% CI (−8.226, −2.432)] and higher experienced discrimination [B = 4.754, *p* = 0.007, 95% CI (1.302, 8.207)] significantly predicted increased COVID-19 peritraumatic distress.ijerph-19-07639-t002_Table 2Table 2Regression results using COVID-19 peritraumatic distress as the criterion.StepVariablesUnstandardizedBeta (SE)95% CI*p*


LLUL
1Gender 




Male (reference)




Female
0.643 (1.583)

−2.471

3.757

0.685

Age




<34 (reference)




35–64
−0.415 (2.836)

−5.995

5.165

0.884

>65
−1.750 (3.137)

−7.922

4.423

0.577

Marital status 




Married/partnered




Other
1.326 (2.492)

−3.576

6.229

0.595

Highest education




College and under (reference)




University degree
−3.184 (1.942)

−7.006

0.638

0.102

Graduate degree
−3.103 (2.129)

−7.292

1.086

0.146

Living arrangement 




Alone (reference)




Friends/others 
6.816 (4.070)

−1.193

14.825

0.095

Family
4.569 (3.181)

−1.689

10.828

0.152

Housing 




House (reference)




Others
1.151 (1.958)

−2.703

5.004

0.557

Years in North America




<15 years (reference)




>15 years
−0.334 (1.587)

−3.457

2.789

0.833

Financial satisfaction 




No (reference)




Neutral
−3.771 (1.906)

−7.522

−0.020

0.049

Yes
−5.184 (2.085)

−9.286

−1.082
**0.013**
Weight gain




Loss (reference)




Same
2.454 (3.031)

−3.511

8.419

0.419

Gain
7.610 (3.230)

1.254

13.967
**0.019**
Health status




Poor (reference)




Good 
−7.231 (1.588)

−10.356

−4.106
**<0.001**
Perceived discrimination




Disagree (reference)




Neutral
6.405 (2.096)

2.282

10.529
**0.002**
Agree
5.852 (2.239)

1.446

10.258
**0.009**
Experienced discrimination




No/others (reference)




Yes
5.152 (1.919)

1.377

8.927
**0.008**2Gender 




Male (reference)




Female
0.263 (1.449)

−2.588

3.113

0.856

Age




<34 (reference)




35–64
−1.749 (2.600)

−6.865

3.367

0.502

>65
−3.788 (2.878)

−9.452

1.875

0.189

Marital status




Married/partnered (reference)




Other
1.083 (2.278)

−3.400

5.565

0.635

Highest education




College and under (reference)




University degree
−2.040 (1.782)

−5.546

1.467

0.253

Graduate degree
−2.265 (1.949)

−6.100

1.570

0.246

Living arrangement 




Alone (reference)




Friends/others 
7.057 (3.719)

−0.262

14.376

0.059

Family
4.115 (2.913)

−1.617

9.848

0.159

Housing 




House (reference)




Others
0.923 (1.790)

−2.598

4.445

0.606

Years in North America




<15 years (reference)




>15 years
−0.419 (1.454)

−3.280

2.442

0.773

Financial satisfaction




No (reference)




Neutral
−2.729 (1.749)

−6.170

0.712

0.120

Yes
−3.480 (1.917)

−7.253

0.293

0.070

Weight gain




Loss (reference)




Same
1.016 (2.778)

−4.450

6.482

0.715

Gain
5.447 (2.970)

−0.397

11.291

0.068

Health status




Poor (reference)




Good 
−5.329 (1.472)

−8.226

−2.432
**<0.001**
Perceived discrimination




Disagree (reference)




Neutral
3.636 (1.947)

−0.196

7.467

0.063

Agree
2.254 (2.096)

−1.871

6.378

0.283

Experienced discrimination




No/others (reference)




Yes
4.754 (1.755)

1.302

8.207
**0.007**
Self-contraction worry
−4.340 (1.087)

−6.480

−2.200
**<0.001**
Family-contraction worry
−1.973 (1.076)

−4.082

0.137

0.067
Note: bold *p*-values denote significant effects (*p* < 0.05). CI = confidence interval, SE = standard error, LL = lower limit, UL = upper limit.

### 3.3. Contraction Worry and Loneliness

Self-contraction and family-contraction worry were significantly correlated with loneliness [*r* = −0.396 and *r* = −0.343, respectively (*p*s < 0.001)]. Table 3 presents the results of the two-step hierarchical regression model on loneliness, with self- and family-contraction worry as the primary predictors and the identified potential sociodemographic variables as the covariates. Step 1, with all the sociodemographic covariates, resulted in an *R*^2^ = 0.206, *F* (15, 307) = 5.320, and *p* < 0.001. In Step 2, adding self-contraction worry and family-contraction worry resulted in an *R*^2^ = 0.279, *F* (2, 305) = 15.395, and *p* < 0.001. The difference between the two steps was Δ*R*^2^ = 0.073, *p* < 0.001, suggesting that contraction worry explained a significant proportion of the variance in loneliness. Specifically, self-contraction worry was identified as a significant predictor [B = −0.771, *t* = −2.775, *p* = 0.006, 95% CI (−1.317, −0.224)], but family-contraction worry was not [B = −0.355, *t* = −1.297, *p* = 0.196, 95% CI (−0.894, 0.184)] when controlling for the sociodemographic covariates. Specifically, a 1-point increase in self-contraction worry significantly predicted an increase of 0.77 points in loneliness.

Additionally, among all the sociodemographic variates, poorer health status [B = −1.386, *t* = −3.682, *p* < 0.001, 95% CI (−2.127, −0.645)] and higher perceived discrimination [B = 1.068, *t* = 2.002, *p* = 0.046, 95% CI (0.018, 2.118)] predicted higher loneliness.ijerph-19-07639-t003_Table 3Table 3Regression results using loneliness as the criterion.StepVariablesUnstandardizedBeta (SE)95% CI*p*


LLUL
1Gender 




Male (reference)




Female
0.195 (0.381)

−0.554

0.945

0.608

Marital status 




Married/partnered




Other
−0.215 (0.582)

−1.360

0.930

0.712

Highest education




College and under (reference)




University degree
−0.331 (0.463)

−1.242

0.581

0.476

Graduate degree
−0.312 (0.494)

−1.285

0.660

0.528

Living arrangement 




Alone (reference)




Friends/others 
−0.106 (0.975)

−2.024

1.812

0.913

Family
−0.730 (0.778)

−2.259

0.800

0.349

Housing 




House (reference)




Others
0.454 (0.455)

−0.442

1.351

0.319

Financial satisfaction 




No (reference)




Neutral
−0.950 (0.461)

−1.856

−0.043
**0.040**
Yes
−1.205 (0.741)

−2.190

−0.220
**0.017**
Weight gain




Loss (reference)




Same
1.063 (0.745)

−0.396

2.521

0.153

Gain
1.643 (0.788)

0.092

3.193
**0.038**
Health status




Poor (reference)




Good 
−1.728 (0.388)

−2.492

−0.964
**<0.001**
Perceived discrimination




Disagree (reference)




Neutral
1.372 (0.512)

0.365

2.380
**0.008**
Agree
1.698 (0.545)

0.625

2.770
**0.002**
Experienced discrimination




No/others (reference)




Yes
0.855 (0.468)

−0.067

1.776

0.069
2Gender 




Male (reference)




Female
0.125 (0.365)

−0.592

0.843

0.731

Marital status




Married/partnered (reference)




Other
−0.191 (0.557)

−1.287

0.906

0.732

Highest education




College and under (reference)




University degree
−0.084 (0.446)

−0.961

0.793

0.850

Graduate degree
−0.105 (0.474)

−1.039

0.828

0.824

Living arrangement 




Alone (reference)




Friends/others 
0.037 (0.932)

−1.797

1.872

0.968

Family
−0.810 (0.745)

−2.276

0.656

0.278

Housing 




House (reference)




Others
0.396 (0.436)

−0.462

1.253

0.365

Financial satisfaction




No (reference)




Neutral
−0.796 (0.442)

−1.666

0.073

0.072

Yes
−0.933 (0.481)

−1.880

0.014

0.053

Weight gain




Loss (reference)




Same
0.826 (0.711)

−0.572

2.224

0.246

Gain
1.285 (0.757)

−0.205

2.775

0.091

Health status




Poor (reference)




Good 
−1.386 (0.377)

−2.127

−0.645
**<0.001**
Perceived discrimination




Disagree (reference)




Neutral
0.888 (0.498)

−0.091

1.867

0.075

Agree
1.068 (0.533)

0.018

2.118
**0.046**
Experienced discrimination




No/others (reference)




Yes
0.796 (0.448)

−0.086

1.677

0.077

Self-contraction worry
−0.771 (0.278)

−1.317

−0.224
**0.006**
Family-contraction worry
−0.355 (0.274)

0.196

−0.894

0.184
Note: bold *p*-values denote significant effects (*p* < 0.05). CI = confidence interval, SE = standard error, LL = lower limit, UL = upper limit.

## 4. Discussion

This study aimed to examine whether COVID-19 contraction worry for self and for family would predict COVID-19 peritraumatic distress and loneliness among Chinese residents in North America. It represented a novel approach added to the literature by exploring these relationships in Chinese residents living in North America, and specifically, by examining the possible differential effects of self- and family-contraction worry. In a partial support of the hypotheses, the results identified self-contraction worry as a significant predictor for both COVID-19 peritraumatic distress and loneliness, even after controlling for the related sociodemographic variables (e.g., gender, marital status, education, and housing). Specifically, higher self-contraction worry predicted increased COVID-19 peritraumatic distress and loneliness in Chinese residents in North America.

The results were consistent with previous findings of a positive association between worry and COVID-19 peritraumatic distress [9,10,11]. In the current study, we have extended these results to Chinese residents in North America. As COVID-19 peritraumatic distress is a comprehensive measure of distress incorporating a broad range of mental health syndromes and pathologies, this effect may be explained by the fact that worry was strongly associated with depression, anxiety, arousal, and insomnia [56]. In addition, replicating and expanding on the results of Hoffart et al. [12] and Mayorga et al. [13], the current study revealed that self-contraction worry was significantly associated with loneliness during the COVID-19 pandemic among Chinese residents in North America. Worry may cause increased loneliness by putting a strain on close relationships. Individuals who tend to overthink in ways such as ruminating or worrying may cause friends and family members to become frustrated due to persistent discussion of the issues and unmitigated communion [57]. Unmitigated communion, which is the tendency to assume undue responsibility for the well-being of others, is one of the characteristics of rumination [57]. Ruminators were found to be perceived less favorably by others [58], and they reported more social friction and less emotional support [57].

However, contrary to our hypotheses, family-contraction worry did not significantly predict both COVID-19 peritraumatic distress and loneliness during the pandemic (see Table 2 and Table 3). Despite of a high correlation between self- and family-contraction worry (*r* = 0.751, *p* < 0.001), only self-contraction worry, and not family-contraction worry, significantly predicted COVID-19 peritraumatic distress and loneliness. This was interesting because while considering the collectivistic culture in China [49], we had predicted that family-contraction worry would also matter in psychological well-being. Additionally, a greater proportion of the sample reported that they lived with family (86.1%) compared to those living alone (7.7%) or those living with friends or in other accommodations (5.9%). However, an exploratory one-way ANOVA did not find any significant differences among participants who lived alone, with family, or with friends in self-contraction worry [*F* (2, 878) = 0.153, *p* = 0.858] or family-contraction worry [*F* (2, 878) = 1.141, *p* = 0.320].

Previous research suggested that participants were more anxious regarding their significant others compared to themselves [59], and prior qualitative research found that individuals reported more COVID-19-related worry regarding family members, especially those at a higher risk [23]. However, in this study, family-contraction worry was not significantly associated with COVID-19 peritraumatic distress or loneliness. It is possible that items in the COVID-19 peritraumatic distress and loneliness measures used in this study were both self-oriented, reflecting feelings or emotions related to the self rather than others. For example, a sample item of the CPDI was: “I feel empty and helpless, no matter what I do”; and a sample item of the De-Jong Gierveld Loneliness Scale was: “During the pandemic, I miss having people around me”. Therefore, it was possible that the COVID-19 peritraumatic distress and loneliness scores were only significantly predicted by self-contraction, but not family-contraction worry. Further research is required to test this speculation. In addition, future research may further examine the underlying mechanisms or moderators of the relationship between self-contraction worry and psychological well-being, such as distress and loneliness.

The results also showed that participants with a poorer self-reported health status tended to show higher COVID-19 peritraumatic distress and loneliness. This replicated Yu et al. [44], which identified health status as a risk factor for mental health outcomes (i.e., depression, anxiety, and stress). Participants with pre-existing health conditions were experiencing higher distress, possibly because they were at a higher risk of contracting COVID-19. These participants might have been more detrimentally impacted by the restrictions in social functions and in accessibility of medical services during the pandemic.

Furthermore, the results also identified experienced anti-Chinese discrimination as a risk factor for higher COVID-19 peritraumatic distress (Table 2) and perceived anti-Chinese discrimination as a predictor for higher loneliness (Table 3). These findings were consistent with previous findings that increased experienced discrimination [46] and perceived discrimination [44,48] against Chinese were associated with increased distress or mental health symptoms. Specifically, experienced discrimination was significantly associated with higher levels of COVID-19 peritraumatic distress, suggesting that experiences of discrimination may add a traumatic additional layer to the negative experiences of the pandemic in Chinese residents. In addition, perceived discrimination was significantly associated with loneliness, suggesting those who perceive anti-Chinese discrimination might be more likely to self-isolate and avoid social interactions. This would in turn increase their feelings of loneliness. The results were largely consistent with earlier findings that loneliness served as a mediating factor between perceived discrimination and depression [60] or nonsuicidal self-injury [61]. Together, these findings highlighted the urgency of reducing racial discrimination and alleviating the level of perceived and experienced discrimination against minority groups during the COVID-19 pandemic.

The study had several limitations. Due to the cross-sectional design of the study, it was impossible to infer causality from the findings. Therefore, further research into the causal relationship between contraction worry and COVID-19 peritraumatic distress or loneliness is needed. Additionally, the study was conducted using a community random-sampling procedure, so there might have been limits to the representativeness of the sample and the generalizability of the findings. Finally, the sociodemographic variables were coded into categorical variables, and this may have reduced the sensitivity of the data to capture the relationships between these variables and the outcome variables.

## 5. Conclusions

Despite these limitations, the findings made a novel contribution to the literature by identifying self-contraction worry as a significant risk factor for COVID-19 peritraumatic distress and loneliness in Chinese residents in North America. The results highlighted the negative impact of self-contraction worry on psychological well-being, and also identified some sociodemographic risk factors for distress and loneliness. The results specifically identified the detrimental effects of self-reported health status and discrimination (perceived or experienced) on distress and loneliness during COVID-19 for Chinese residents in North America, a vulnerable population. The COVID-19 pandemic’s impacts on mental health may have short-term and long-term mental health consequences, and therefore require updating policy plans and mental health resources to best support minority populations [62,63]. The findings may have implications for social and health services, and the results suggested the need for interventions that aim to reduce contraction worry, break racial discrimination, and mitigate the psychological impacts of the pandemic in minority populations. Interventions should also target the promotion of community engagement that fosters support and trust. More programs are needed to improve coping skills, and resilience may help to reduce distress and loneliness related to the COVID-19 pandemic [30,36].

## Figures and Tables

**Table 1 ijerph-19-07639-t001:** Sample characteristics and their relationships with COVID-19 peritraumatic distress and loneliness (*N* = 943).

Variables		COVID-19 Peritraumatic Distress			Loneliness		
	*N* (%)	Mean (*SD*)	*F*	*p*	Mean (*SD*)	*F*	*p*
Gender			5.114	**0.024**		3.880	**0.049**
Male	319(33.8%)	49.075 (14.106)			17.433 (3.428)		
Female	608(64.5%)	51.445 (14.764)			17.929 (3.488)		
Age group			4.695	**0.009**		0.940	0.391
<34	115 (12.2%)	54.408 (16.706)			18.126 (3.841)		
35–64	653 (69.2%)	50.378 (14.341)			17.767 (3.537)		
>65	175 (18.6%)	48.777 (13.576)			17.510 (2.926)		
Marital status			6.352	**0.012**		4.411	**0.036**
Married/partnered	775 (82.2%)	49.971 (14.170)			17.647 (3.466)		
Other	164 (17.4%)	53.294 (16.143)			18.317 (3.463)		
Highest education			5.504	**0.004**		2.802	**0.061**
College and under	215 (22.8%)	52.991 (14.843)			18.111 (3.252)		
University degree	407 (43.2%)	50.746 (15.037)			17.876 (3.606)		
Graduate degree	320 (33.9%)	48.596 (13.478)			17.397 (3.412)		
Employment			1.674	0.196		0.695	0.405
Employed/self-employed	528 (56.0%)	49.961 (14.799)			17.676 (3.628)		
Other	414 (43.9%)	51.253 (14.214)			17.876 (3.249)		
Living arrangement			5.550	**0.004**		6.007	**0.003**
Alone	73 (7.7%)	49.642 (14.650)			18.853 (3.554)		
Friends/others	56 (5.9%)	57.362 (13.303)			18.844 (3.483)		
Family	812 (86.1%)	50.201 (14.521)			17.608 (3.438)		
Children under			0.338	0.561		0.851	0.357
Yes	410 (40.2%)	50.149 (14.326)			17.624 (3.672)		
No	572 (56.1%)	50.736 (14.788)			17.848 (3.329)		
Housing			8.312	**0.004**		15.643	**<0.001**
House	741 (78.6%)	49.843 (14.518)			17.528 (3.428)		
Other	200 (21.2%)	53.363 (14.362)			18.698 (3.495)		
Religion			0.278	0.757		0.103	0.903
No	564 (59.8%)	50.193 (13.922)			17.801 (3.346)		
Christian/Catholicism	227 (24.1%)	50.951 (16.043)			17.757 (3.777)		
Other	147 (15.6%)	50.952 (14.678)			17.646 (3.463)		
Pet			1.968	0.162		2.265	0.133
Yes	71 (6.9%)	46.980 (13.816)			17.079 (3.625)		
No	291 (28.6%)	49.811 (14.629)			17.818 (3.459)		
Years in North America			3.020	**0.083**		0.889	0.346
<15 years	454 (48.1%)	51.449 (14.305)			17.891 (3.487)		
>15 years	480 (50.9%)	49.724 (14.712)			17.665 (3.454)		
Financial satisfaction			35.069	**<0.001**		30.703	**<0.001**
No	230 (24.4%)	56.811 (15.102)			19.227 (3.737)		
Neutral	390 (41.4%)	50.460 (14.106)			17.695 (3.172)		
Yes	322 (34.1%)	46.247 (13.040)			16.833 (3.283)		
Weight change			5.921	**0.003**		3.372	**0.036**
Loss	23 (2.4%)	46.389 (12.930)			16.529 (4.017)		
Same	238 (25.2%)	47.674 (13.205)			17.472 (3.261)		
Gain	99 (10.5%)	53.573 (16.712)			18.413 (3.826)		
Health status			102.254	**<0.001**		93.624	**<0.001**
Poor	355 (37.6%)	56.817 (15.484)			19.2278 (3.436)		
good (>3)	588 (62.4%)	46.971 (12.690)			16.945 (3.211)		
Experienced discrimination			55.410	**<0.001**		40.223	**<0.001**
No	723 (76.7%)	48.593 (13.363)			17.361 (3.332)		
Yes	220 (23.3%)	57.108 (16.405)			19.125 (3.587)		
Perceived discrimination			25.013	**<0.001**		25.706	**<0.001**
Disagree	156 (17.40)	43.527 (12.762)			16.144 (3.182)		
Neutral	398 (44.30)	50.949 (13.739)			17.751 (3.172)		
Agree	344 (38.30)	53.197 (15.145)			18.511 (3.654)		

Note: bold *p*-values (*p* < 0.10) refer to the variables that were entered in the regression models in Tables 2 and 3. SD = standard deviation.

## Data Availability

The SPSS data, syntax, and output files can be found at https://doi.org/10.17605/OSF.IO/E354C (accessed on 19 June 2022).

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
