# Peer review of "COVID-19 Peritraumatic Distress and Loneliness in Chinese Residents in North America: The Role of Contraction Worry"

_ijerph, 2022, doi:10.3390/ijerph19137639_

Round 1

Reviewer 1 Report

Dear Authors

After reviewing your manuscript I would like to expose some points that, It think is needed to explain deeply:

1.- Firstly I have some doubts about the possible bias that includes in your study the inclusion of Chinese international students (people who have not lived in North America and follow the customs of another country) and peolple that live in North America and will foreseeably have another cultural approach.

Furthermore, Is there a minimum length of residence to be included in the study? Is an international student who has just arrived in North America also eligible?

Could you explain how these elements have been analyzed or included in your study?

2.-It has been impossible for me to find any reference to this study having been evaluated and approved by the ethics committee of any university or research center.

Please can you include this information in the material and methods section?

3.- The authors include the statistical analyses performed as part of the results and not as part of the Data Analysis and methods section.

Please change this as it does not fit the format of the journal.

4.- Can you explain why you use a significance level of 0.1 and not 0.05, which is more accepted in the health sciences environment?

5.- You have used the ANOVA test assuming that your population conforms to normality but I have not been able to find anywhere in the study these data to test for normality and homoscedasticity of the sample. Nor have you indicated it in the text.

Please, could you provide information that would allow us to know if parametric tests can be used?

6.- The authors include the limitations of the study in conclusions when this should be included in the discussion.

Reviewer 2 Report

This is interesting paper focused on the possible predictors of loneliness during the COVID pandemic. The concept of loneliness is very  relevant for mental health issues, since it can have different detrimental effects. I think that the paper is well-written and focused on an specific aspect domain (loneliness) which has not been adequately studied so far.

However, I would suggest authors to make some minor changes in order to further improve the quality of the manuscript:

1. in the introduction, authors should specify the reasons for considering the COVID pandemic as a new traumatic stressor associated with a higher risk for developing mental health issues. Authors should consider to quote some relevant papers such as Unützer J, Kimmel RJ, Snowden M. Psychiatry in the age of COVID-19. World Psychiatry. 2020 Jun;19(2):130-131. doi: 10.1002/wps.20766. PMID: 32394549; PMCID: PMC7214943; Bridgland VME, Moeck EK, Green DM, Swain TL, Nayda DM, Matson LA, Hutchison NP, Takarangi MKT. Why the COVID-19 pandemic is a traumatic stressor. PLoS One. 2021 Jan 11;16(1):e0240146. doi: 10.1371/journal.pone.0240146. PMID: 33428630; PMCID: PMC7799777.

2. Authors should comment on the fact that during the pandemic it has been witnessed an increase in social isolation, looking at papers such as Holt-Lunstad J. A pandemic of social isolation? World Psychiatry. 2021 Feb;20(1):55-56. 

3. In the discussion, authors should compare their findings with a similar paper focused on loneliness conducted in Europe (e.g., Sampogna G, Giallonardo V, Del Vecchio V, Luciano M, Albert U, Carmassi C, Carrà G, Cirulli F, Dell'Osso B, Menculini G, Belvederi Murri M, Pompili M, Sani G, Volpe U, Bianchini V, Fiorillo A. Loneliness in Young Adults During the First Wave of COVID-19 Lockdown: Results From the Multicentric COMET Study. Front Psychiatry. 2021 Dec 10;12:788139. doi: 10.3389/fpsyt.2021.788139. PMID: 34955932; PMCID: PMC8703162.)

4. Have you controlled your analyses for the effect of age group? Did you find any difference?

5. The subheadings of paragraphs 3.2 and 3.3 are not very clear. Please rephrase it.

6. In the final part of the conclusions, authors should discuss some possible practical implications for mental health professionals, looking at some relevant papers published by international scientific associations such as WPA and EPA (i.e., Stewart DE, Appelbaum PS. COVID-19 and psychiatrists' responsibilities: a WPA position paper. World Psychiatry. 2020 Oct;19(3):406-407. doi: 10.1002/wps.20803. PMID: 32931089; PMCID: PMC7491607; Kuzman MR, Curkovic M, Wasserman D. Principles of mental health care during the COVID-19 pandemic. Eur Psychiatry. 2020 May 20;63(1):e45. doi: 10.1192/j.eurpsy.2020.54. PMID: 32431255; PMCID: PMC7267055; McDaid D. Viewpoint: Investing in strategies to support mental health recovery from the COVID-19 pandemic. Eur Psychiatry. 2021 Apr 26;64(1):e32. doi: 10.1192/j.eurpsy.2021.28. PMID: 33971992; PMCID: PMC8134893.)

Round 2

Reviewer 1 Report

Dear authors

Thank you for the inclusion of information that will facilitate the understanding of your article to future readers.

kind regards